# Cell Line Platforms Support Research into Arthropod Immunity

**DOI:** 10.3390/insects12080738

**Published:** 2021-08-17

**Authors:** Cynthia L. Goodman, David S. Kang, David Stanley

**Affiliations:** Biological Control of Insects Research Laboratory, USDA/Agricultural Research Service, 1503 S. Providence Rd., Columbia, MO 65203, USA; cindy.goodman@usda.gov (C.L.G.); david.stanley@usda.gov (D.S.)

**Keywords:** innate immunity, antimicrobial peptide, RNAi, lysozyme, pathogen, signaling pathway, IMD, Toll, hemocyte, eicosanoid, antiviral

## Abstract

**Simple Summary:**

Many insect and tick species are serious pests, because insects damage crop plants and, along with ticks, transmit a wide range of human and animal diseases. One way of controlling these pests is by impairing their immune system, which protects them from bacterial, fungal, and viral infections. An important tool for studying immunity is using long-lasting cell cultures, known as cell lines. These lines can be frozen and thawed at will to be used in automated tests, and they provide consistent results over years. Questions that can be asked using cell lines include: How do insects or ticks recognize when they have been infected and by what organism? What kinds of defensive strategies do they use to contain or kill infectious agents? This article reviews research with insect or tick cell lines to answer these questions, as well as other questions relating to immunity. This review also discusses future research strategies for working with cell lines.

**Abstract:**

Innate immune responses are essential to maintaining insect and tick health and are the primary defense against pathogenic viruses, bacteria, and fungi. Cell line research is a powerful method for understanding how invertebrates mount defenses against pathogenic organisms and testing hypotheses on how these responses occur. In particular, immortal arthropod cell lines are valuable tools, providing a tractable, high-throughput, cost-effective, and consistent platform to investigate the mechanisms underpinning insect and tick immune responses. The research results inform the controls of medically and agriculturally important insects and ticks. This review presents several examples of how cell lines have facilitated research into multiple aspects of the invertebrate immune response to pathogens and other foreign agents, as well as comments on possible future research directions in these robust systems.

## 1. Introduction

Invertebrate innate immunity is a generalized reaction that does not depend on prior immune experiences. It has been reviewed from several perspectives, such as the model fruit fly, *Drosophila melanogaster* [1], prophenoloxidase activation [2], and eicosanoid signaling [3,4]. Here, we review the topic with respect to using insect and tick cell lines as research platforms for investigating innate immunity (Figure 1). Since invertebrate immunity has been thoroughly reviewed elsewhere, we begin with a brief overview, then turn to our topic.

Invertebrates express immune responses to invasions by living and nonliving agents (such as injected fluorescent-labeled beads or implanted cuticles). In general, invertebrates express innate, but not adaptive, immune reactions to invaders, while vertebrates express innate and adaptive antibody-based immunity. Innate immunity is a generalized reaction that does not depend on prior immune experiences. Nonetheless, the idea that insects can express adaptive immunity in the form of an immune memory probably emerged in the early 1980s with a study on specific immune memories in male American cockroaches, *Periplaneta americana* [5]. Recent studies have continued advancing the concept of immune memory in insects [6,7,8,9], and we generally foresee broad recognition of a form of adaptive immunity, albeit without antibodies, in invertebrates.

Although there is considerable overlap, invertebrate immunity is assorted into humoral and cellular immunity. Cellular immunity is launched almost immediately when an infection is detected, and it is responsible for clearing over 90% of the infecting bacteria from the hemolymph circulation within the first 2 h post-infection (PI; [10]) (Please see Table 1 for definitions of the frequently used abbreviations). Infections are detected via pattern recognition receptors (PRRs) [11,12,13], which activate pathogen-specific signaling pathways [9,13,14]: the Toll pathway (for Gram-positive bacteria and fungi), the Immune deficiency (IMD) pathway (Gram-negative), with some exceptions in both cases, and the Janus Kinase and Signal Transducer and Activator of Transcription (JAK/STAT) pathway [15].

Larger invaders, such as parasitoids, are encapsulated by several layers of hemocytes, which also become melanized and connected to internal structures [16]. These actions follow hemocyte migration toward the sites of infections and wounds via chemical gradients [17] for immune defense actions.

Humoral immune reactions are recorded as the presence of antimicrobial peptides (AMPs) specific to Gram-positive and Gram-negative bacteria and to invading fungi in the hemolymphs of infected insects about 6–12 h PI [18,19]. Insects also express potent antiviral mechanisms [7,9,20,21] (Figure 1).

Cell culture-based systems, using either continuously replicating insect or tick cell lines or primary cell cultures, are widely used in research programs. Immortal cell lines have been employed in studies involving virus–host interactions, virus propagation, recombinant protein production, hormone function, insecticide mode-of-action and screening programs, and most recently, food production [22,23,24,25,26]. The use of short-term primary cell cultures (which can survive days to months) predate the use of cell lines; they have been valuable tools in many aspects of physiological research, including immunity [3,27,28,29,30,31]. Established insect cell lines have advantages over short-term culture studies in that, once established, cell lines can be cryopreserved for decades, are amenable to high-throughput studies, are less labor-intensive (therefore, more cost-effective), and lead to consistent results. Here, we focus on immortal cell lines from agriculturally and medically important insects or ticks to investigate different aspects of the immune response. Our goal is to highlight selected studies as examples of the use of cell lines as tools in this research area. It is not an exhaustive approach, e.g., we will not reference *D.*
*melanogaster* studies, as they have been recently reviewed [20,32,33]. Our target audience includes seasoned investigators of invertebrate immunity, as well as those recently drawn into this area. Thus, we included more citations than necessary as a guide to those recently drawn into immune-related research.

## 2. Pathogen-Associated Molecular Patterns (PAMPs), Pattern Recognition Receptors (PRRs), and Opsonins

Invertebrate cell lines have been used to identify pathogen-associated molecular patterns (PAMPS) (Table 2). Ha Lee et al. [34] reported that bacterial peptidoglycans (PGN) were more potent activators of the antimicrobial protein (AMP) gene cecropin B (CecB) than lipopolysaccharides (LPS) in the *Bombyx mori* cell line NISES-BoMo-Cam1. They found that PGN from *Escherichia coli* stimulated the expression of several antibacterial peptide genes and other genes, whereas PGN from *Micrococcus luteus* activated a few genes. They showed that *E. coli* PGN or cells elicited a higher expression of the peptidoglycan recognition protein gene involved in the prophenoloxidase activation pathway compared to *M. luteus* PGN or cells. Using the mosquito cell line C6/36, Mizutani et al. [35] reported the constitutive expression of low levels of two AMPs, cecropin and defensin, which were upregulated by exposure to lipopolysaccharides.

Shaw et al. [36] found that two lipids isolated from bacteria-infected cells, 1-palmitoyl-2-oleoyl-snglycero-3-phosphoglycerol (POPG) and 1-palmitoyl-2-oleoyl diacylglycerol (PODAG), stimulated the IMD pathway in ISE6 tick cells (Table 3). They also reported that exposing the cells to these lipids protected the tick cells from infection by two rickettsia-related bacteria, *Anaplasma phagocytophilum* and *A. marginale*.

**Table 2 insects-12-00738-t002:** Examples of the insect cell lines used in immune-related studies.

Order	Species of Origin	Stage/Tissue of Origin	Cell Line Designation	Research Focus	References
Coleoptera	*Anthonomus grandis*	Embryo	BRL-AG-1	Humoral Responses	[37]
	*A. grandis*	Embryo	BRL-AG-3A	Humoral Responses	[37]
	*A. grandis*	Embryo	BRL-AG-3C	Humoral Responses	[37]
	*Tribolium castaneum*	Pupa/Adult	BCIRL-TcA-CLG1	Signaling Pathways	[38]
Diptera	*Aedes aegypti*	Neonate larva	Aag-2	Signaling Pathways, Cellular Responses, Humoral Responses	[37,39,40,41,42,43,44,45,46,47,48]
	*Ae. aegypti*	Neonate larva	AF5 and subline AF319	Signaling Pathways	[43,45]
	*Ae. aegypti*	Neonate larva	ATC-10 (CCL-125)	Signaling Pathways, Humoral Responses	[42,45,49,50]
	*Aedes albopictus*	Neonate larva	C6/36	PAMPs, Signaling Pathways, Humoral Responses	[35,42,44,45,46,51,52,53,54,55,56,57]
	*Ae. albopictus*	Neonate larva	C6/36 HT	Signaling Pathways	[39]
	*Ae. albopictus*	Neonate larva	C7-10	Signaling Pathways, Cellular Responses, Humoral Responses	[37,42,45,58,59,60]
	*Ae. albopictus*	Neonate larva	U4.4	Signaling Pathways	[42,44,45,61]
	*Anopheles* *gambiae*	Neonate larva	4a-2	Humoral Responses	[37]
	*A. gambiae*	Neonate larva	4a-3A	Humoral Responses	[37,62]
	*A. gambiae*	Neonate larva	4a-3B	Humoral Responses	[37]
	*A. gambiae*	Neonate larva	Sua1B	Signaling Pathways, Cellular Responses, Humoral Responses	[37,62,63]
	*A. gambiae*	Neonate larva	Sua5.1*	Opsonins, Signaling Pathways	[64]
	*Anopheles stephensi*	1st Instar larva	4a-3A	Signaling Pathways	[62]
	*A. stephensi*	1st Instar larva	4a-3B	Signaling Pathways	[65]
	*A. stephensi*	1st Instar larva	LSTM-AS-43 (MSQ43)	Signaling Pathways, Humoral Responses	[62]
	*Culex quinquefasciatus*	Ovary (adult)	Hsu	Signaling Pathways	[56]
	*Culex tarsalis*	Embryo	CT	Signaling Pathways	[56]
	*Lutzomyia longipalpis*	Embryo	LL5	Signaling Pathways	[66,67]
	*Sarcophaga peregrina*	Embryo	NIH-Sape-4	Humoral Responses	[37,68]
Hemiptera	*Anasa tristis*	Embryo	BCIRL-AtE-CLG15A	Signaling Pathways	[38]
Lepidoptera	*Opodiphthera (Antheraea) eucalypti*	Pupal ovaries	Ae	Cellular Responses	[69]
	*Bombyx mori*	Ovary (larval)	Bm5	Signaling Pathways	[55,70,71]
	*B. mori*	Ovary	BmN	Signaling Pathways	[72]
	*B. mori*	Ovary	BmN4	Signaling Pathways	[55]
	*B. mori*	Ovary (larval)	BmN-SWU1	Signaling Pathways	[73]
	*B. mori*	Ovary	NISES-BoMo-Cam1	PAMPs, Signaling Pathways	[34]
	*Choristoneura* *fumiferana*	Midgut (larval)	IPRI-CF-1	Signaling Pathways	[74]
	*C. fumiferana*	Midgut	FPMI-CF-203	Signaling Pathways	[55]
	*Chrysodeixis (Pseudoplusia) includens*	Embryo	UGA-CiE1	Cellular Responses, Humoral Responses	[75]
	*Estigmene acraea*	Hemocyte (larval)	BTI-EA-1174-A	Cellular Responses, Humoral Responses	[37,76]
	*Helicoverpa zea*	Ovary (pupal)	BCIRL-HzAM1	Signaling Pathways, Humoral Responses	[77,78,79,80]
	*H. zea*	Midgut (larval)	RP-HzGUT-AW1	Signaling Pathways	[55]
	*Helithis virescens*	Ovary (pupal)	BCIRL-HvAM1	Signaling Pathways	[38]
	*Lymantria dispar*	Ovary (pupal)	IPLB-Ld-652Y	Cellular Responses	[81]
	*Malacosoma disstria*	Hemocyte (larval)	IPRI-Md-66	Signaling Pathways, Cellular Responses, Humoral Responses	[82]
	*M. disstria*	Hemocyte (larval)	IPRI-Md-108	Signaling Pathways, Cellular Responses, Humoral Responses	[82]
	*Manduca sexta*	Embryo	MRRL-CHE-20	Signaling Pathways	[74]
	*Perina nuda*	Ovary (pupal)	NTU-Pn-HH	Cellular Responses	[81]
	*Plodia interpunctella*	Unspecified	KSU-P15.3	Signaling Pathways	[74]
	*Spodoptera exigua*	Embryo/neonate larva	Se301	Cellular Responses	[69]
	*S. exigua*	Hemocyte (larval)	SeHe920-1a	Cellular Responses	[69]
	*S. exigua*	Neonate larva	UCR-Se-1	Cellular Responses	[81]
	*Spodoptera frugiperda*	Ovary (pupal)	IPLB-Sf-5-5C	Cellular Responses	[81]
	*S. frugiperda*	Ovary (pupal)	Sf21	Signaling Pathways	[55]
	*S. frugiperda*	Ovary (pupal)	Sf9	Signaling Pathways, Humoral Responses	[83,84,85,86,87]
	*S. frugiperda*	Ovary (pupal)	Sf9-SF (serum-free)	Signaling Pathways	[55]
	*Spodoptera littoralis*	Ovary	Sl2	Signaling Pathways	[55]
	*Spodoptera litura*	Ovary (pupal)	IBL-Sl-1A	Cellular Responses	[81]
	*Trichoplusia ni*	Embryo	High Five (BTI-TN-5B1-4)	Signaling Pathways, Cellular Responses	[55,69,71,87,88]
	*T. ni*	Embryo	High Five-SF (serum-free)	Signaling Pathways, Humoral Responses	[55]

A key attribute of arthropod innate immunity is to recognize foreign threats. Pattern recognition receptors (PRRs) are a class of receptors that recognize PAMPs [11,12,13]. Consequently, PRRs initiate the intracellular signaling cascades central to the organism’s defense to pathogenic incursions [9,14]. *D. melanogaster* cell lines have been primarily used for the identification and study of pattern recognition receptors (PRRs) [13,15]. Moita et al. [64] investigated the role of integrins (specifically, the Arg.Gly.Asp-recognizing receptors) in phagocytosis using cell line 5.1* from *Anopheles gambiae*. Growing these cells in suspension elevated their ability to engulf bacteria compared to the attached cultures, suggesting that the adhesion and phagocytosis processes share receptors. This helped identify a new integrin gene involved in phagocytosis, BINT2. Their study confirmed that the glycoprotein TEP1 (thiol-ester motif-containing protein-1) is not a phagocytic receptor but an opsonin that indirectly promotes phagocytosis.

## 3. Signaling Pathways and Signaling Molecules Involved in Antimicrobial and Antiviral Responses

### 3.1. Phagocytosis Related Signaling Pathways

The recognition of foreign PAMPs activates several signaling pathways that stimulate cellular and humoral responses. Mizutani et al. [35] investigated a phagocytosis signaling pathway in the mosquito cell line C6/36 using fluorescein-labeled spheres or bacteria. They reported that exposing the cells to the JNK-specific inhibitor SP600125 led to reduced sphere/bacteria uptake. Similarly, incubating cells with the same inhibitor led to a reduced accumulation of acridine orange, as well as the uptake of the West Nile virus. Hence, the JAK/STAT signaling pathway acts in phagocytosis, endocytosis, and virus entry in mosquito cells.

Trujillo-Ocampo [39] used the cell lines C6/36 HT from *Aedes albopictus* and Aag-2 from *Ae. aegypti* to investigate the 14-3-3ε and 14-3-3ζ protein actions in phagocytosis. The 14-3-3 proteins interact with protein partners as adapters, activators, and repressors, and they are involved in regulating signaling pathways and other cellular processes. Using RNAi to decrease protein expression in Aag-2 cells, they recorded changes in the cytoskeleton organization and decreased phagosome maturation and phagocytosis.

### 3.2. Signaling Pathways Associated with Antimicrobial Humoral Responses

Using the mosquito cell lines Sua1B and 4a3a and MSQ43, combined with RNAi, Luna et al. [62] showed that the expression of two antimicrobial peptide (AMP) genes gambicin (*gam1*) and defensin (*def1*) were regulated by the IMD pathway. They determined that overexpression of the NF-κB transcription factors involved in the IMD pathway (designated as Relish 2 or REL2) and the Toll pathway (designated as REL1) stimulated the expression of cecropin (*cec1*), *gam1*, and *def1*. These findings suggest there is crosstalk between the Toll and IMD pathways within the cells.

Barletta et al., 2012 [40] incubated Asg-2 cells with a variety of immune stimuli, including Gram-positive or Gram-negative heat-inactivated bacteria, fungal zymosan, or the Sindbis virus, and used qPCR to quantify the expression of genes specific in the Toll, IMD, and JAK/STAT pathways. Gram-positive/negative bacteria and zymosan stimulated the expression of key markers of both the Toll (cactus) and IMD (REL2) pathways. For the JAK/STAT pathway, bacteria increased the expression of a thiol-ester motif-containing protein, TEP, and the virus upregulated expression of the transcription factor STAT in Aag-2 cells.

Zhang et al. [41] exposed the Aag2 cell line to a variety of microbes, including Gram-negative and Gram-positive bacteria and fungi, and found an upregulation of a number of AMPS (including three defensins, six cecropins, and gambicin). They identified the specific pathways involved in AMP induction. Most AMP genes activated by the Gram-negative bacterium were regulated by the IMD pathway. Gambicin was controlled by the combination of all three pathways (IMD, Toll, and JAK/STAT). Their findings differed from *D. melanogaster*, which regulate AMP expression primarily through the Toll and IMD pathways.

The embryonic cell line LL5 was used to investigate the Toll and IMD pathways in the sandfly *Lutzomyia longipalpis* [66]. The investigators silenced the repressor genes for the Toll and IMD pathways (*cactus* or *dorsal* for Toll and *caspar* or *relish* for IMD), then exposed the cell line to heat-killed bacteria, yeast, or live protozoa. The cells exhibited increased expression of AMP genes after each pathway repressor was silenced. After the cells were incubated with *E. coli*, the authors noted increases in the mRNAs encoding *cactus, caspar*, *cecropin*, and *defensin 2*, (but not *attacin*) and decreases in the mRNAs encoding *dorsal* and *relish*. Similar findings followed exposure to two other bacteria and a yeast. Likewise, live protozoa challenges led to the upregulation of *cactus* with no change in *caspar* and increased the expression in both the *dorsal* and *relish* levels.

The Toll pathway was studied using the lepidopteran cell line BmN-SWU1, as well as the dipteran cell line S2 [73]. The S2 cells were co-transfected with vectors from the Toll-interleukin-1 receptor domains from the *Bombyx mori* Toll family members, which showed that BmToll11 and BmToll9–1 can activate the *drosomycin* and *diptericin* promoters. The overexpression of the Toll-interleukin-1 receptor domains in *B. mori*, BmN-SWU1, cells resulted in the upregulation of a variety of AMPs, and their silencing led to the inhibition of AMP expression. In vivo experiments confirmed that BmToll11, BmToll9–1, and five Spätzle genes were upregulated in *B. mori* larvae after an infection by *E. coli* and *Staphylococcus aureus*. One of the *BmSpz* genes, *BmSpz2*, interacted with BmToll11 and BmToll9–1. This study provided evidence that *Toll* and *Spz* act in the *B. mori* innate immunity.

Shaw et al. [36] reported that the x-linked inhibitor of the apoptosis protein (XIAP), which discourages rickettsia bacterium infections, is linked to the IMD pathway in a tick cell line. They also showed that Bendless directly interacts with XIAP. Silencing the *uev1a* and *bendless* genes led to an increased bacterial load. The authors conducted a detailed phylogenic analysis, from which they proposed two functionally distinct IMD pathways in ticks and insects.

Rao et al., 2011 [83] described how the activities of the AMP gene promoters from different species regulate the expression of these genes in a species-specific manner in S2 and Sf9 cells (from *Spodoptera frugiperda)*. They suggested that transcription complexes using common nuclear factors are combined with species-related coregulators and that these are responsible for the species-specific regulation of AMP gene expression.

Muller et al. [65] reported that the 4a-3B line from *A. gambiae* constitutively expressed six pathway (PPO) genes. LaPointe et al. [82] reported that two lepidopteran cell lines derived from hemocytes Md-66 and Md-108 generated intracellular melanin in response to the bacterium *Bacillus subtilis*. The cells did not release PO into the medium during the early stages of infection, but longer incubations led to the formation of a melanotic coagulum around the granular-like cells.

Braconid wasps transmit immunosuppressive bracoviruses when they parasitize their hosts. Beck and Strand [88] used the lepidopteran cell line High Five to elucidate the mechanism of the *Microplitis demolitor* bracovirus. They reported that a conditioned medium from virus-infected High Five cells blocked the melanization of bacteria-challenged *Manduca sexta* plasma. After the *egf1.0/1.5* (a putative melanization inhibitor based on a sequence analysis) expression was silenced, the conditioned medium no longer inhibited the melanization. Additionally, High Five cells were used to determine the virus viability. The bracovirus viability declined in the presence of PO but was unaffected when Egf1.0 was present. The authors concluded that activated PO is directly responsible for reducing the virus and parasitoid viability.

A functional PO cascade occurred in a mosquito cell line U4.4 from *Ae. albopictus* [61]. Exposing the cells to *E. coli* or the arbovirus Semliki Forest virus (SFV) led to an increased medium melanization, which was correlated with a reduced virus viability. They produced a recombinant SFV that expressed the PO pathway blocker Egf1.0 and noted that its expression enhanced the spread of the virus. U4.4 cells are morphologically similar to oenocytoids, the primary source of PO in mosquito plasma.

### 3.3. Eicosanoid-Related Signaling

Prostaglandins (PGs), a group within eicosanoids, mediate the immune responses and other physiological processes in arthropods [3,4]. PGs are synthesized in vertebrates by cyclooxygenases, although, in insects, the enzymes have been recently identified as peroxidases [97,98]. Barbosa and Rebello [51] reported that prostaglandin A1 (PGA_1_) mediates the synthesis of stress proteins during cells’ lag phase (specifically, HSPs 27, 29, 70, 80, and 87 kDa) in the C6/36 *Ae. albopictus* cell line. When the cells were in the exponential and stationary phases, PGA_1_ induced fewer HSPs and in lower quantities. Similarly, de Meneses et al. [52] confirmed PGA_1_ mediates the increased HSP synthesis and reported that PGA_1_ also increases the synthesis of HSPs 23 and 57. These two HSPs are primarily regulated by PGA_1_, not heat shock. Treating the cells with an inhibitor of cyclooxygenase, aspirin, did not influence the HSP70 levels when the cells were maintained at their standard growth temperature (28 °C) but upregulated their synthesis at a higher temperature of 37 °C. HSPs act in a variety of immunoregulatory roles at the intracellular and extracellular levels [99,100].

Later studies showed that a variety of PGs influence the up- and downregulation of proteins of differing functionalities, including those involved in defensive responses [77,78]. The lepidopteran cell line BCIRL-HzAM1 was treated with either PGA_1_ or PGE_1_, followed by a proteomic and bioinformatic analysis [77]. The authors reported changes in the expression of 34 proteins, with functionalities involving the protein action, lipid metabolism, signal transduction, protection, cell functions, and metabolism. Of these, significant changes in the HSP levels were influenced by both PGs, including the up- and downregulation of the HSP70 levels. The expression of the proteins involved in cell defense included the antioxidants superoxide dismutases (Mn and Cu/Zn) and glutathione-S-transferase. These enzymes act in the cellular host defense against reactive oxygen species, which are often elevated as part of the innate immune response [101]. A follow-up study working with the two-series PGs was performed using the same cell line [78]. The incubation with PGA_2_ influenced the expression of 60 proteins, whereas the PGE_2_ and PGF_2a_ treatments influenced a few proteins. The expression of the antioxidant proteins were altered—specifically, thiol peroxiredoxin; glutathione S-transferase; and the heat shock-related proteins (including heat shock cognate 70, HSP20.7, and HSP60). Other immune-related proteins affected included the growth blocking peptide-binding protein and lipopolysaccharide-binding protein.

PGs also influence the post-translational modification of selected proteins [79]. The HzAM1 cells were again exposed to PGs (PGA_2_, PGE_1_, or PGF_2α_) for shorter time periods (20–40 min). Significant modifications in the phosphorylation levels occurred in 31 proteins, with decreased levels in 15, increased levels in another 15, and one protein with either increased or decreased phosphorylation, depending on the specific PG treatment. Changes in the phosphorylation of five HSPs were recorded (e.g., HSP60, HSP70, and HSP90), as well as that of two proteins that regulate HSP activities (Hsc-70-interacting-protein and DnaJ homolog shv).

Johnson and Howard [74] recorded the impacts of selected eicosanoid synthesis inhibitors on the responses of three lepidopteran cell lines (MRRL-CHE-20 from *M. sexta*, IPRI-CF-1 from *Choristoneura fumiferana*, and KSU-P15.3 from *Plodia interpunctella*) to the *Bacillus thuringiensis* cryproteins, CryIA(c) and CryIC. They documented the influence of the inhibitors on the cell viability. This was similar to Li et al. [38], who reported that PG synthesis inhibitors, as well as PGs themselves, lead to either reduced cell numbers or cell death in a concentration-dependent manner in cell lines from three insect orders. Johnson and Howard [74] reported that the inhibitors that decreased the cell viability lessened the toxicity of the endotoxins. They performed in vivo experiments with these compounds and found that they were nontoxic to larvae and that the lipoxygenase inhibitor, nordihydroguaiaretic acid, decreased the Cry1Ac toxicity. The authors suggested that the antioxidant activities, potentially involving eicosanoids, are part of the insect responses to endotoxins.

Burlandy et al. [53] showed the involvement of PGA_1_ on virus replication in the mosquito cell line C6/36. They reported a dose-dependent reduction of the vesicular stomatitis virus (VSV) in this cell line, recording up to a 95% decrease with 8-μg PGA_1_/mL. PG-treated cells increased the expression of HSP70, although VSV plus PG-treated cells exhibited reduced HSP70 levels. These studies strongly support the findings that PGs act in insect immunity at the whole-animal level [3,4].

## 4. Antiviral Signaling Pathways

One of the major insect RNA-based responses to viral challenges is the induction of the small interfering RNA (siRNA) pathway [9,71]. The RNAi pathway is activated when dsRNAs from the virus replication cycle are shortened into siRNAs by the enzyme Dicer2 (Dcr2). These siRNAs are incorporated into an RNA-Induced Silencing Complex (RISC) and unwound, creating the guide to target RNA. The protein Argonaute (Ago2 and/or Ago3), a component of RISC, then selects a virus-derived small interfering RNA (vsiRNA) to serve as a guide RNA to target complimentary viral RNAs, resulting in their degradation. There are other, small RNA (sRNA) pathways, all evolved to protect cells from invading viruses [86,102,103,104].

### 4.1. sRNA-Related Pathways

Hoa et al. [63] identified RNAi-related genes in *An. gambiae*, such as those that encode dicer and argonaute-like proteins, and then knocked down their expression in the Sua1B mosquito cell line. They recorded a reduction in RNAi activity and confirmed that the expression of the Dcr2, Ago2, and Ago3 proteins were required by this pathway.

Sigle and McGraw [45] reviewed the genes that are upregulated in response to virus infections in mosquitos, including RNAs of unknown functions [42,43,44]. Ruckert et al. [56] characterized the sRNA responses of mosquito cell lines from three viruses (the arbovirus, West Nile virus (WNV), and two insect-specific viruses: the flavivirus Calbertado virus and the rhabdovirus Merida virus) to determine whether the vpiRNAs are involved in the antiviral responses. They reported that the sRNA responses differed among the host cell–virus combinations. The authors also noted that the *Culex quinquefasciatus* Hsu cell line generated vpiRNAs when infected with the Merida virus and concluded that the major sRNA response of the *Culex* mosquito cell lines to WNV infection is via the RNAi pathway involving exogenous siRNAs.

Pitaluga et al. [67] challenged the sandfly cell line LL5 with virus-like particles (VLPs) from the West Nile virus (WNV). Their study showed that dsRNA stimulated an antiviral response in the sandfly cells, noting that a transfection with ssRNA (either sense or antisense) blocked VLP infection in this cell line.

Swevers et al. [55] investigated the impact of persistent RNA virus infection with Flock house virus (FHV) and Macula-like virus (MLV) on RNAi efficiency. They found that MLV can infect all the cell lines tested and that FHV infection was less common. Virus-free Sf21 cells and FHV-free High Five cells (with low levels of MLV) were used to evaluate the absence/presence of a virus on RNAi gene silencing in these cell lines, which showed that the RNAi machinery was not inhibited in persistently infected cells. The RNAi inhibitor genes FHV (*B2*) and MLV (*VSR*) were intact in persistently infected cell lines, and they suggested that the virus levels in persistently infected cell lines are too low to influence the RNAi process in lepidopteran cell lines. Santos et al. [71] compared the influence of RNAi components in acute versus persistent viral infections in High Five cells. They overexpressed key RNAi proteins (e.g., Dcr2 and Ago2) and recorded the increased cell defenses against acute infection by the Cricket Paralysis Virus (CrPV), with no impact on the transcript levels of the persistently infecting Macula-like Latent Virus (MLV). Working with the Bm5 cell line, they found virus-specific small RNAs associated with Ago2. The investigators knocked down *dcr2* and *ago2*, which led to increased MLV transcript levels, although an overexpression of these enzymes did not alter the virus titers. The authors concluded that the siRNA pathway is involved in the antiviral response in acute and persistent viral infections in lepidopteran cells.

The RNAi antiviral response is involved in lepidopteran cell lines infected with baculoviruses, important biocontrol agents. Mehrabadi et al. [84,85] reported that *Autographa californica* nucleopolyhedrovirus (AcMNPV) modifies the microRNA expression in Sf9 cells and that the viral *p35* gene inhibits the RNAi antiviral pathway in these cells. Karamipour et al. [86] studied the response and role of the siRNA components Dcr2 and Ago2 to the AcMNPV infection in Sf9 cells. The Dcr2 and Ago2 transcript levels were increased after the virus challenge. When gene-specific dsRNAs were introduced, reductions in the transcript levels followed, with a concomitant increase in the AcMNPV viral genome titers. Similarly, the overexpression of the RNAi suppressor p35 elevated the virus levels, confirming that the siRNA pathway was activated following the baculovirus challenge to act in the antidefense of the Sf9 cells.

Using tick ISE6 cells, Garcia et al. [95] tested three viral suppressors of RNAi (NS1 protein (Influenza virus), NSs (Tospovirus Tomato spotted wilt virus), and HC-Pro (Zucchini yellow mosaic virus)). Their results suggested that NS1, which binds to RNA, inhibits RNAi in tick cells by sequestering the siRNAs. NS1 also partially inhibited the RNAi activity.

### 4.2. Non-RNA-Related Antiviral Signaling Pathways

In Liu et al. [46], two mosquito cell lines from *Aedes* spp. were infected with the dengue virus, and the expression of the selected signaling molecules were knocked down to determine their involvement in the antivirus response. Knocking down the Extracellular Receptor Kinase (ERK) led to increased virus titers in the cells, whereas the knocking down of Mitogen-Activated Protein Kinases (MAPKs), JNK, and p38 had no effect. Liu et al., concluded that Ras/ERK signaling, which regulates the AMP levels, act in the mosquito cell defense against dengue virus infection. Similar results were also reported by Xu et al. [105]. The JAK/STAT pathway, coupled with insulin, acts in defending insect cells against West Nile virus [106].

Russell et al. [47] developed unique RT-qPCR and luciferase reporter assays to investigate the induction of IMD and Toll pathways in the mosquito cell line Aag-2, focusing on genes upregulated in vivo in *Ae**. aegypti* by the activation of NF-κB signaling, including REL1A (Toll pathway) or REL2 (IMD pathway). The Toll pathway was not induced by challenging the cells with fungi, bacteria, or viruses. The IMD, but not the Toll, pathway detected the presence of a general arbovirus PAMP (dsRNA mimic poly(I:C)) and was induced by the insect-specific cricket paralysis virus.

Liu et al. [70] engineered a Bm5 cell line to permanently express the Toll-related receptor gene that encodes BmToll9-1. The authors reported that BmToll9-1 increased the expression of Dicer2 (RNAi) and selected the transcription factors (depending on the PAMP) involved in the JAK/STAT and Toll pathways. They reported that activating this Toll-related receptor via LPS reduced the induction of AMP effector genes and IMD or JAK/STAT pathway genes. The authors did not find a BmToll9-1 receptor that directly interacted with the endogenous RNAi machinery.

### 4.3. Other Uses of Cell Lines in Viral Pathway Studies

Cell lines can also serve as valuable tools indirectly in investigating immune-related pathways. Souza-Neto et al. [54] reported that the JAK/STAT pathway acts in defense against the dengue virus. They used the mosquito cell line C6/36 to quantify the viral titers via the plaque assay to determine the effects of silencing pathway-related genes on viral titers in a tissue-specific manner, as well as for identifying two JAK-STAT pathway-regulated and infection-responsive dengue virus restriction factors (DVRFs). Engineered cell lines were used to express the pathway players, such as in the expression of the suppressor of the cytokine signaling 2 (BmSOCS2) gene in the *B. mori* cell line BmN to confirm its antiviral properties [72].

## 5. Antimicrobial Cellular Responses

A range of hemocyte types have been identified by different researchers, usually associated with specific orders. Examples of the hemocyte types described in dipterans, hymenopterans, and lepidopters include plasmatocytes, granulocytes, (also known as granular cells; the most abundant), prohemocytes, plasmatocytes, oenocytoids, coagulocytes, crystal cells, spherulocytes, and thrombocytoids [16,28,29,30,107]. Each cell type has its own role in the immune responses (some of which are not fully understood). Hemocytes exhibit a variety of defensive responses to foreign agents, including phagocytosis, nodulation, and encapsulation. Phagocytosis has been investigated in established cell lines. Other mechanisms, such hemocyte activation, migration, aggregation, and spreading, have been extensively investigated in primary cultures [3,27,28,30].

### Phagocytosis

In Fallon and Sun’s review [37], they described three lines with phagocytotic properties (two dipteran, Sua lB and mbn-2, and one lepidopteran, BTI-EA-1174-A) and also reported that the *Ae. albopictus* C7-10 cell line phagocytized fluorescent-labeled bacteria. Similarly, Mizutani et al. [35] recorded the uptake of labeled, heat-killed *E. coli* in the C6/36 mosquito cell line. Barletta et al. [40] incubated Aag-2 cells (generated from *Ae. aegypti* neonates) with fluorescent latex beads and found that, after 1 h, virtually all the cells had phagocytized beads. Trujillo-Ocampo et al. [39] confirmed phagocytosis in Aag-2 cells and described the formation of phagolysosomes.

Wittwer et al. [108] described the phagocytosis of FITC-labeled beads by a hemocyte cell line from *Estigmene acraea*, BTI-EA-1174-A. The phagocytosis was enhanced by the addition of bacterial LPS. Yang et al. [81] compared the phagocytotic properties of five lepidopteran cell lines (IPLB-Sf-5-5C, UCR-Se-1, IBL-Sl-1A, NTU-Pn-HH, and IPLB-Ld-652Y), reporting that Sf-5-5C expressed the highest phagocytosis while the Pn-HH cells were unresponsive. Three hemocyte types were described from the Sf-5-5C cells (non-basophilic cells, light basophilic cells, and basophilic cells), with the first type responsible for phagocytosis. Three of the cell lines phagocytized the baculoviruses to which they were susceptible, suggesting that phagocytosis is not only an immune response but may be involved in viral entry. Chisa et al. [69] developed a cell line from *Spodoptera exigua* hemocytes, SeHe920-1a, from which they isolated subclones. All the lines exhibited varying levels of phagocytic activity. They also tested three non-hemocyte cell lines (derived from *Antheraea eucalypti, Trichoplusia ni*, and *S. exigua*) that phagocytized the particles to a much lower degree. Johnson et al. [75] characterized the *Chrysodeixis (Pseudoplusia) includens* embryonic cell line UGA-CiE1, reporting that these cells morphologically resembled granulocytes and were recognized by anti-granulocyte monoclonal antibodies and phagocytized fluorescently labeled *E. coli* and *M. luteus*. Similarly, LaPointe et al. [82] generated two cell lines from *Malacosoma disstria* hemocytes, Md-66 and Md-108, which bound and phagocytized selected bacteria. These lines displayed granulocyte-like and plasmatocyte-like properties based on morphological and molecular characterization methods (including monoclonal antibody typing).

Tick cell lines exhibit phagocytosis. Examples include the cell lines IDE12 (from *Ixodes scapularis*) and DAE15 (from *Dermacentor andersoni*), which phagocytized the heat-killed and viable Lyme disease spirochete *Borrelia burdorfi* with or without the presence of an endosymbiont. The phagocytosis was very efficient in 70–90% of cells, depending on the treatment [91]. Only 1% of cells from another *I. scapularis* cell line, ISE6, exhibited phagocytotic behavior in this study. Another tick cell line, BME26, generated from the embryos of *Rhipicephalus (Boophilus) microplus* phagocytized yeast [96]. Kurtti and Keyhani [89] reported the phagocytosis of a fungus by two tick cell lines. The phagocytosis did not inhibit the fungal growth, and the authors suggested that phagocytosis, in this case, represented an alternative strategy for pathogen invasion. Teixeira et al. 2016 reported on phagocytosis in eight cell lines from selected tick species using flow cytometry and confocal or immunofluorescence microscopy to more accurately distinguish between the internalized *Borrelia*
*burgdorferi* and bacteria attached to the cell membranes. The authors concluded that the tissue origin of the cell line may influence its association with spirochetes and subsequent phagocytic activities.

## 6. Antimicrobial and Antiviral Humoral Responses

The humoral response involves the synthesis and release of AMPs and other chemicals into the hemolymph from hemocytes or fat body cells. AMPs typically appear in hemolymph of infected insects about 6–12 h after infections. Other chemicals include lysozymes, oxygen and nitrogen-free radicals, stress-related proteins, and enzyme cascades, including the PPO pathway [33,41,99,100]. Humoral responses have been activated against bacterial, fungal, and viral infections.

### 6.1. Antimicrobial Peptides (AMPS)

Matsuyama and Natori (1988) [68] described the synthesis of three AMPs by a dipteran line, NIH-Sape-4, in response to *E. coli* exposure. Fallon and Sun (2001) [37] reported thirteen cell lines that released antimicrobial agents, with many studies describing fully or partially characterized AMPs. The peptides included cecropins, defensins, attacins, and diptericins [37]. Sun and Fallon (2002) [58] characterized the cecropin genes from the *Ae. albopictus* line C7-10. The expression of AMP genes have been described in other mosquito cell lines, including lines from *A. gambia* and *A. stephensi* (Sua1B, 4a3a, and MSQ43) [62]. In response to a bacterial challenge, these lines expressed the *defensin A (Def1)* and the mosquito-specific *gambicin (Gam1)* AMP genes. Likewise, the C6/36 cell line from *Ae**. albopictus* expressed the *cecropin A1* and *cecropin B* genes when exposed to the bacterium *Francisella tularensis* spp. *novicida* (the causal agent of tularemia) [57]. Zhang et al. [41] reported the production of RNAs encoding cecropins, defensins, and gambicin after incubating Aag2 cells for 12 h with a variety of bacteria, with only a limited response to the fungus *Candida albicans*.

The lepidopteran cell lines Sf9 and High Five express AMPs in response to bacteria [87], although this study did not conclusively identify them. Johnson et al. [75] reported a cell line from the soybean looper *C. includens*, UGA-CiE1, that upregulated the expression of the AMP lebosin (*Pi-leb*) following a challenge by the bacterium *M. luteus*.

Tick cell lines express AMPs either constitutively or when challenged by microbes. The cattle tick cell line BME26 constitutively expressed the AMP peptide genes encoding microplusin and defensin [96]. Tonk et al. [93] identified defensin genes in the cell line IRE/CTVM19 from *Ixodes ricinus* and showed that these genes differed phylogenetically and structurally. They also predicted that these peptides would be active against Gram-positive bacteria.

### 6.2. Lysozymes

Immune-responsive lysozymes are expressed in numerous invertebrate cell lines. Gao and Fallon [48] reported that Aag-2 cells secreted a lysozyme after being challenged by heat-killed bacteria. This protein shared a 50% amino acid identity with lysozymes from two *Anopheles* species. A lysozyme was also produced by an *Ae**. albopictus* cell line, C7-10, exposed to heat-killed bacteria, with sequence similarities to the protein from Aag-2 cells [59]. Nasr and Fallon [60] showed that the C7-10 lysozyme was more effective against the Gram-positive bacterium *M. luteus* than the Gram-negative bacterium *E. coli*.

Wittwer et al. [76] isolated a lysozyme from the lepidopteran hemocyte cell line, BTI-EA-1174-A, and showed that it was upregulated in response to lipopolysaccharides from bacteria. Similarly, the *M. disstria* hemocyte cell line (Md-66) expressed a lysozyme after long-term incubation with *B. subtilis*. However, another *M. disstria* cell line (Md-108) did not respond to a bacterial challenge with lysozyme production [82]. On the other hand, Johnson et al. [75] showed that CiE1 cells constitutively expressed an immune-related lysozyme.

Lysozymes are also produced by microbial-infected tick cell lines. The DAE100 cell line, derived from *D. andersoni* embryos, responded to incubation with heat-treated *E. coli* by increasing the transcripts encoding a lysozyme. From the sequence analysis, this lysozyme had the highest homology to the arthropod c-type lysozymes [92].

### 6.3. Other Immune-Related Proteins

Invertebrate cells express a considerable range of agents, including iron transfer proteins, heat shock proteins, detoxification enzymes, and other stress-related proteins. The tick cell line BME26 has phagocytic properties and expresses AMPs. In this study, Esteves et al. [96] identified eleven immune-related transcripts expressed by these cells, including ferritin, serine proteases, protease inhibitors, a heat shock protein, glutathione S-transferase, peroxidase, and NADPH oxidase. This highly expressed ferritin is involved in a nutritional immunity response in that it sequesters iron from bacteria and other microorganisms, depriving these organisms of needed nutrients [109]. Geiser et al. [49] confirmed this response in the mosquito cell line CCL-125. These cells were exposed to a heat-inactivated mixture of bacteria (*B. subtilis* and *E. coli*) in the presence of iron and responded by upregulating the expression of intracellular and extracellular ferritin.

Turning to viral defense, Colpitts et al. [50] overexpressed the genes that were downregulated during a virus infection and found that they caused a significant reduction in the virus infection in CCL-125 mosquito cells. They identified one of these proteins as a pupal cuticle protein (PCP) and a matrix metalloprotease (MMP). An additional work suggested the PCP binds viral proteins and inhibits West Nile virus entry, whereas MMP indirectly influences viral infections, although the action mechanism is unclear.

Weisheit et al. [94] performed a transcriptomic and proteomic analysis of two tick cell lines, IDE8 (from *I. scapularis*) and IRE/CTVM19 (from *I. ricinus*), that were either infected or mock-infected by the tick-borne encephalitis virus (TBEV). They identified fourteen transcripts that potentially act in the immune-related responses, including phagocytosis, the complement system, the ubiquitin–proteasome pathway, protein folding, and the piRNA pathway. They found that knocking down HSP90 and HSP70 led to increased viral loads in IDE8 cells. The authors suggested these results may have been due to an impaired RNAi response. However, a study using the lepidopteran cell line BCIRL-HzAM1 and the *Helicoverpa zea* Single Nucleocapsid Nuclear Polyhedrosis Virus (HzSNPV) found that virus replication required an upregulation of *hsp70* expression [80]. These response differences are likely due to the biological differences between the two systems. Weisheit et al. [94] also identified the proteases trypsin and longipain. Trypsin acts in several immune signaling pathways, and the cysteine protease longipain is involved in anti-parasite responses [110]. CD36, a class B scavenger receptor, is upregulated during bacterial infections. The authors concluded that RNAi is not the only mechanism involved in tick cell antiviral responses.

## 7. Conclusions

Continuously replicating arthropod cell lines from dipterans, lepidopterans, and ticks are effective tools for investigating innate immunity. They have been used to characterize pattern recognition receptors, explore the interactions of these receptors with pathogens and the signaling pathways they activate, and identify the defensive responses of cells to specific foreign agents. Future works will certainly include taking advantage of newly developed/optimized state-of-the-art techniques, such as single-cell analysis and genome editing. Single-cell analysis will enable researchers to identify cell type-specific immune responses [111,112], especially given that many invertebrate cell lines consist of heterogenous populations of cells. Molecular biologists are continuing to expand the genome editing techniques that will allow investigators to acquire an even deeper understanding of the functions and interactions of selected genes [113,114], with RNA editing also playing an important part in this constantly developing technology [115].

## Figures and Tables

**Figure 1 insects-12-00738-f001:**
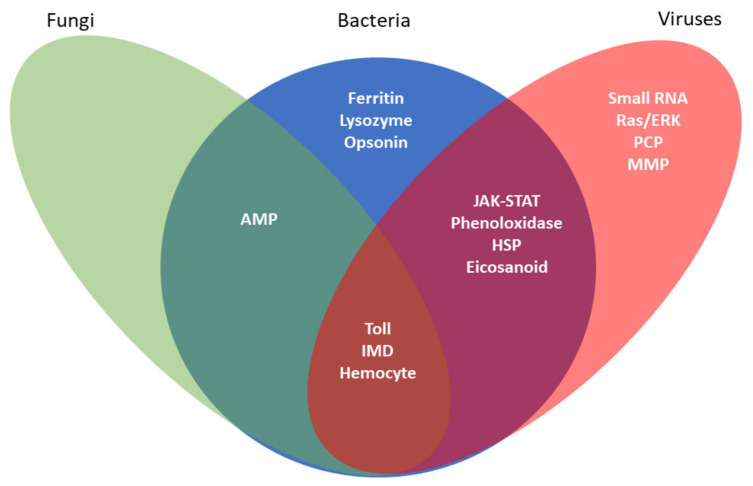
Invertebrate cell lines are amenable for studies of innate immunity. Immortalized cell lines have proven to be an invaluable tool for investigating insects’ varied humoral and cellular immune responses. This Venn diagram illustrates the broad array of immune responses studied in insect and tick cell lines and the pathogens that elicit these defensive responses. Each oval represents the immune response associated with a specific type of pathogen. Overlapping regions represent immune responses elicited by multiple pathogenic stimuli.

**Table 1 insects-12-00738-t001:** Definitions of the reoccurring abbreviations.

Term	Abbreviation
anti-microbial peptides	AMPs
Argonaute	Ago
Dicer2	Dcr2
Double-stranded RNA	dsRNA
Extracellular receptor kinase	ERK
Heat shock protein	HSP
Immune deficiency	IMD
Janus Kinase and Signal Transducer and Activator of Transcription	JAK/STAT
Jun N-terminal Kinase	JNK
Lipopolysaccharides	LPS
Mitogen-Activated Protein Kinase	MAPK
Pathogen-associated molecular pattern	PAMP
Pattern recognition receptor	PRRs
Peptidoglycan	PGN
Phenoloxidase or prophenoloxidase	PO or PPO
Piwi RNA	piRNA
Prostaglandin	PG
Post-infection	PI
Relish	REL
small interfering RNA	siRNA
Single-stranded RNA	ssRNA
RNA Interference	RNAi
Small RNA	sRNA
thiol-ester motif-containing protein	TEP
viral-derived piwi-associated RNA	vpiRNA
viral small interfering RNAs	vsiRNA

**Table 3 insects-12-00738-t003:** Examples of the tick cell lines used in immune-related studies.

Order	Species of Origin	Stage/Tissue of Origin	Cell Line Designation	Research Focus	References
Parasitiformes	*Amblyomma americanum*	Embryo	AAE2	Cellular Responses	[89]
	*Amblyomma variegatum*	Molting larva	AVL/CTVM17	Cellular Responses	[90]
	*Dermacentor andersoni*	Embryo	DAE15	Cellular Responses	[91,92]
	*D. andersoni*	Embryo	DAE100	Humoral Responses	[92]
	*Hyalomma anatolicum*	Embryo	HAE/CTVM8	Cellular Responses	[90]
	*Ixodes ricinus*	Embryo	IRE/CTVM19	Cellular Responses	[90,93,94]
	*Ixodes scapularis*	Embryo	IDE8	Cellular Responses, Humoral Responses	[90,94]
	*I. scapularis*	Embryo	IDE12	Cellular Responses	[89,91]
	*I. scapularis*	Embryo	ISE6	PAMPs, Signaling Pathways, Cellular Responses	[36,90,91,95]
	*Ixodes ricinus*	Embryo	IRE/CTVM19	Humoral Responses	[93,94]
	*Rhipicephalus* *appendiculatus*	Molting nymph	RA243	Cellular Responses	[90]
	*R. appendiculatus*	Embryo	RAE/CTVM1	Cellular Responses	[90]
	*Rhipicephalus (Boophilus) microplus*	Embryo	BME/CTVM2	Cellular Responses	[90]
	*R. microplus*	Embryo	BME26	Cellular Responses, Humoral Responses	[96]

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
