# Peer review of "Cell Line Platforms Support Research into Arthropod Immunity"

_insects, 2021, doi:10.3390/insects12080738_

Round 1

Reviewer 1 Report

This paper reviews important achievements in insect immunity using cellular platforms. It is of great value for scientists to understand the cellular immunity and humoral immunity of insects at the level of the cell. The review organized well and I have only a few suggestions for the authors.

  • About the title, most of information is about Insecta, cannot represent arthropod.
  • I suggest the reference that firstly report the cell line could be list in table 2 for respect.
  • “3. Pattern Recognition Receptors (PRRs) and Opsonins”

More information about types of PRRs are recommended. And the response pathways can be briefly informed to introduce the follow-up architecture.

  • I will recommend combine “6.1. Phagocytosis” and “1. Phagocytosis Related Signaling Pathways” as main content of “Antimicrobial Cellular Responses”. Integrating the two parts is more convenient for the readers and avoid repetition.
  • Phagocytosis is the most studied process in culture cells, it is worth introducing in detail. Besides, the audiences probably will be interest with other initial immune response to pathogen invaded by membrane fusion, insertion of infection apparatus, etc.
  • “7. Antimicrobial and Antiviral Humoral Responses” more likely to be “Molecular involved in humoral responses”. I suggest combine it with “Signaling Pathways Associated with Antimicrobial Humoral Responses” as content of “Antimicrobial Humoral Responses”
  • It is suggested to increase schematic diagram of the general technical route and research methods for immune research by using cell platform.

Author Response

The first point suggests most information in our review is about insects and not about other arthropods. We mention ticks in several areas of the manuscript and include a range of citations to immune-related research using established tick cell lines. We feel our manuscript did not bring the tick work into sufficient focus, partly due to an error in an event within our word processor. Specifically, we accidently  reduced Table 3 to one line in the final manuscript and we did not catch this mistake before submission. Thus, in the revision we include the missing third table meant to highlight the tick work. In view of our new table, we prefer the term arthropod, or at times invertebrate, meant to include the substantial work on established tick cell lines.

The second point suggests we cite the first report on insect cell lines. With due respect for the pioneers in establishing permanently replicating insect cell lines, we cite Smagghe et al, 2009 (Citation #22), which reviews the history in insect cell lines and cites all the early work. We prefer to not repeat those early citations in this work.

The third point suggests adding more information on PRRs and Opsonins. We agree and make the point by combining sections 2 & 3 to highlight their importance.  We also expand on the role PRRs play in the immune process in this new section.

The fourth point suggests we combine sections 6.1 and 4.1. Here we respond also to Point 7 which suggests we combine two other sections. We are disappointed with these suggestions as the reviewer is in effect asking us to completely restructure our manuscript. We are disappointed because we began our writing process by developing a design and structure meant to guide our development of a cohesive and reasonably complete manuscript that conveys a meaningful story. We feel we have accomplished our goal and prefer to leave the structure of the manuscript in our original design.

In addressing two of the reviewer’s suggestions, we skipped the suggestion that phagocytosis “is the most studied process in culture cells and it is worth introducing in detail”. The reviewer is correct for work with primary cell lines, but this statement does not apply to established cell lines. That is why in our introductory comments, beginning on line 88, we limit our focus to established, immortal cell lines:  “Here, we focus on immortal cell lines from agriculturally and medically important insects or ticks to investigate aspects of the immune response”.

Finally, the reviewer suggests we increase our schematic diagram to include “the general technical route and research methods for immune research…”. We feel two comments are germane. One, we designed our diagram to convey a conceptual appreciation of the broad range of microbes that infect insects and the complex arsenal of immune responses insects evolved to handle the infections. Two, approaching research methods in insect immunity has been addressed to greater and lesser extents in broader reviews of insect immunity, taken generally, and is not necessary to the intent of our current manuscript. For a single example, in their review of eicosanoid signaling in insect immunity, Stanley and Kim (2014, not cited in the current manuscript) include a section entitled “Getting it done: a brief section on methods”.

Stanley, D., Kim, Y. (2014) Eicosanoid signaling in insects: from discovery to plant protection. Critical Reviews in Plant Sciences 33: 20-63.  

Again, we thank Reviewer #1 for meaningful comments that led us to think carefully about our manuscript and led to improvements. We hope the manuscript quality has been sufficiently improved to merit acceptance. If the situation is otherwise, we are ready to continue our work as necessary.

Reviewer 2 Report

This paper 'Cell line platforms support research into arthropod immunity' by Goodman et al. is a well written, researched and comprehensive review of the topic. Most reviews on the topic are focused on Drosophila and mosquitoes, and I'm certain this review will benefit a larger community of arthropod immunologists. This paper has tackled the different components of the innate immune mechanisms and the further break down is helpful too. 

Author Response

We thank the reviewers for their encouragement and their work on our first submission, which has helped us develop an improved manuscript. Because Reviewer #2 did not indicate a need for specific improvements, we focused on responding to Reviewer #1.